# Spatial Patterns of Key Villages and Towns of Rural Tourism in China and Their Influencing Factors

Qiuyu Zou [1], Jianwei Sun [1,*], Jing Luo [2], Jiaxing Cui [2] and Xuesong Kong [3]

1 School of Geographical and Environmental Sciences, Guizhou Normal University, Guiyang 550025, China; zouqiuyu@gznu.edu.cn
2 Hubei Provincial Key Laboratory for Geographical Process Analysis and Simulation, Central China Normal University, Wuhan 430079, China; luojing@mail.ccnu.edu.cn (J.L.); cuijiaxing@ccnu.edu.cn (J.C.)
3 School of Resource and Environmental Sciences, Wuhan University, Wuhan 430079, China; xuesongk@whu.edu.cn
* Correspondence: sunjianwei@gznu.edu.cn

**Abstract:** This study takes 1597 key villages and towns part of rural tourism in China as research objects and uses the ArcGIS spatial analysis method, combined with the geodetector and the multi-scale geographically weighted regression model, to analyze the intensity and spatial differentiation of factors influencing the spatial distribution of these villages and towns. (1) The key villages and towns of rural tourism exhibit clustering distribution patterns with more locations in the east than in the west. The center of gravity of this distribution shifts to the northwest, displaying a "belt–point" trend with scattered hot spots in Beijing, Tianjin, Hebei, and the Yangtze River Delta, while cold spots are primarily concentrated in the northwest. (2) Spatial variation in the key villages and towns of rural tourism is due to multiple factors, of which population density, resident consumption expenditure, and per capita GDP display the highest explanatory powers for the spatial distribution of tourism. (3) The primary influencing factors are spatial differences in their roles and substantial local imbalances. Positive and negative correlation analysis units exhibit the aggregation characteristics of being blocked and banded. These results can provide valuable guidance for the development of rural tourism, promoting its sustainable development and contributing to the revitalization of rural areas.

**Keywords:** key villages of rural tourism; key towns for rural tourism; spatial distribution; geodetector; multiscale geographically weighted regression

## 1. Introduction

Tourism developed rapidly in China due to the establishment of various economic reforms by the government [1]; moreover, rural tourism has attracted much attention in recent years from all walks of life as a new form of tourism [2]. Rural tourism plays an important role in promoting rural economic development and agricultural transformation and upgrading, as well as increasing the income and wealth of farmers [3]. Compared with traditional urban tourism, rural tourism emphasizes a combination of rural natural and humanistic landscape resources, integrating various elements of tourism such as leisure and entertainment, ecological sightseeing, and agricultural experience, and demonstrates the unique charm and humanistic style of the countryside [4]. However, under the dual promotion of policy and market, on the one hand, rural tourism increases the momentum of vigorous development [5]. However, issues such as severe homogenization, lagging tourism infrastructure, unreasonable land use, and the destruction of resources and the environment remain restricting the sustainable development of rural tourism [6]. In this context, China's rural tourism is entering a stage of quality and characteristic development. Key villages of rural tourism refer to villages relying on the natural and humanistic landscape resources of the countryside, integrating rural culture, leisure and vacation, eco-tourism, agricultural experience, etc. [4]. They are selected, recommended, and finally recognized by the Chinese

Ministry of Culture and Tourism. To cooperate with the implementation of the national rural revitalization strategy, the Ministry of Culture and Tourism has announced four batches of 1399 key villages and two batches of 198 key towns for rural tourism since 2019. In addition, the construction of key villages for rural tourism was included in the Law of the People's Republic of China on Rural Revitalization Promotion in April 2021. The construction of key villages and towns for rural tourism plays an important role in promoting rural economic and social development. Therefore, exploring the overall spatial layout of key villages and towns in rural tourism in China and elucidating the influencing factors of the spatial distribution of key villages and towns on the basis of the realities of rural tourism are necessary initiatives [7]. A systematic study of the spatial patterns and influencing factors of rural tourism in key villages and towns is of great significance for enriching the research content of rural geography and revitalization.

At present, studies on rural tourism conducted locally and abroad are becoming increasingly abundant. In the 1980s, rural tourism began to flourish in foreign countries [8], which mainly included the concept of rural tourism [9], a driving mechanism [10], the rural development model [11], and the impact of rural tourism [12,13]. Since then, the academic community has gradually explored rural tourism image perception [14], its spatial layout [15], and the influencing factors of spatial distribution [16]. When examining issues related to rural tourism, a few relatively simple methods for data collection have been initially used, such as interviews, questionnaires, and comparative analysis. However, as research progresses, scholars use increasingly sophisticated empirical research and econometric analysis methods for quantitative analysis, such as constructing mathematical models [17], geographical information system (GIS) spatial techniques [18], and spatial econometric models [8]. Therefore, foreign scholars have investigated the rural tourism space and have used interdisciplinary disciplines to improve their theoretical systems. Eventually, they employed GIS technology and mathematical methods to analyze tourism space quantitatively, which has laid an important theoretical foundation for research on the spatial structure of rural tourism [19]. In recent decades, domestic rural tourism has rapidly developed and studied from a wide range of perspectives, mainly focusing on the concept of rural tourism [20], development paths [21,22], the alleviation of tourism poverty [23], and studies on spatial structure [24]. The impetus for the booming development of domestic rural tourism comes mainly from the demand and supply sides. On the demand side, urban residents want to escape from fast-paced life and are dissatisfied with crowded mass tourism. On the supply side, the rural environment and traditional culture have become the resource base for rural tourism [2]. Additionally, the field of geography has increasingly taken advantage of comprehensive spatiotemporal analysis to conduct research on the spatial distribution and influencing factors of rural tourism sites [25]. These research objects include various sites of rural tourism, such as villages with minority characteristics [26], forest villages [27], the key villages of rural tourism [5], and beautiful leisure villages [7]. Meanwhile, research scales have involved national and provincial areas [19,28], while the research methods used have covered spatial analysis such as the nearest neighbor index, kernel density analysis, the geographic concentration index, imbalance index, and spatial statistical methods [29] to portray the spatial distribution characteristics of villages. Simultaneously, geographic probes [30], geographically weighted regression [28], and grid analysis [25] have been used to explore the influencing factors of village distribution and their differences so as to better understand the regularity of village spatial distribution. The factors influencing spatial distribution include natural and human geographic factors, such as topography, climate, water systems, transportation, population, and economy [25,31]. From a comprehensive point of view, although current academic research on the spatial distribution of different types of tourist sites and influencing factors is relatively mature, research on the evolution of the spatial pattern of key villages and towns of rural tourism in a large-scale pattern and its influencing factors has not yet attracted sufficient attention. First of all, the majority of existing studies have highlighted the spatial distribution characteristics of rural tourism key villages on the national scale from a static perspective.

However, few studies have explored the spatial patterns of four batches of key villages and two batches of key towns in rural tourism from a dynamic perspective. Moreover, studies that investigate the influencing factors and the spatial distribution of key villages and towns of rural tourism using a combination of geographic probes and multiscale geographically weighted regression (MGWR) methods are lacking.

In summary, the study of key villages in rural tourism focuses on spatial structure, spatial distribution characteristics, sustainable development strategies, and the influencing factors of spatial distribution. This study uses ArcGIS spatial analysis, geographic probes, and the MGWR model to analyze the spatial evolution pattern of key villages and towns of rural tourism systematically and the factors that influence them on a national level. In this manner, the status of key villages in rural tourism can be better assessed, thus consolidating the results of poverty eradication and promoting rural economic development and improvements in the livelihoods of the people. This study can provide a scientific reference and practical basis for the systematic assessment and identification of key villages and towns for rural tourism as well as rational protection and development.

## 2. Data and Methods

### 2.1. Data Sources and Processing

This study obtained the list of key villages and towns related to rural tourism in China from the official website of the State People's Committee (http://www.seac.gov.cn/, accessed on 21 March 2022), which designates 1399 key villages in 4 batches and 198 key towns in 2 batches for a total of 1597 key areas. The vector data of these villages and towns were obtained from the Baidu coordinate picker (http://api.map.baidu.com/, accessed on 2 April 2022). All maps presented in this paper are based on the standard map service of the National Bureau of Surveying and Mapping Geographic Information Services for download (http://bzdt.ch.mnr.gov.cn/, accessed on 26 March 2022. Furthermore, data on a digital elevation model of China with a resolution of 30 m and on climate and river water systems were obtained from the Geospatial Data Cloud (http://www.gscloud.cn/, accessed on 6 August 2022). The socioeconomic data of provinces (autonomous regions and cities) and prefecture-level cities are from the China Statistical Yearbook 2021 and the China City Statistical Yearbook, and some data are from the statistical yearbooks of domestic provinces. The study area was mainland China (excluding Hong Kong, Macao, and Taiwan).

### 2.2. Research Methodology

To explore the spatial pattern characteristics of key villages and towns of rural tourism in China and its influencing factors, first, the nearest neighbor index method was used to discriminate the type of spatial distribution experienced by key villages and towns. Then, we employed the Tyson polygon coefficient of variation (CV) to test the accuracy of the spatial type. Second, the standard deviation ellipse (SDE) was used to reveal the spatial distribution profile and dominant direction of key villages and towns, followed by the kernel density estimation (KDE) method to portray the high- and low-density key villages and towns of rural tourism. Finally, in terms of influencing factors, we used geographic probes to analyze the degree of influence of each indicator on the distribution of villages and towns, followed by the MGWR model to examine the mutual work and spatial heterogeneity of each indicator further. In this manner, we can more comprehensively understand the spatial distribution patterns and influencing factors of the villages and towns of rural tourism sites.

### 2.2.1. Nearest Neighbor Index

The closest proximity index was used to analyze the mutual proximity of key tourism villages and towns and determine their spatial distribution type [6]. Equation (1) presents the calculation formula as follows:

$$R = \frac{\overline{r_1}}{\overline{r_E}} = \frac{1}{2}\sqrt{n/A} \times \overline{r_1}, \tag{1}$$

where $\overline{r_1}$ is the actual nearest distance, $\overline{r_E}$ is the theoretical nearest distance, $A$ is the area of the region, and $n$ is the number of key villages and towns of rural tourism in the study area. $R = 1$, $R > 1$, and $R < 1$ signify that the distribution type of villages is random, uniform, and agglomeration, respectively.

### 2.2.2. Standard Deviational Ellipse

The SDE is an effective spatial statistical method that can accurately reveal the overall characteristics and the spatial distribution of geographical elements. It is primarily used to describe the contours of spatial distribution and the dominant directions of key villages and towns in rural tourism [32]. Equations (2) and (3) provide the calculation formulas:

$$SDE_x = \sqrt{\sum_{i=1}^{n}(x_i - \overline{X})^2 / n} \text{ and} \tag{2}$$

$$SDE_y = \sqrt{\sum_{i=1}^{n}(y_i - \overline{Y})^2 / n} \tag{3}$$

where $SDE_x$ and $SDE_y$ represent the long and short axes of the ellipse, respectively; $x_i$ and $y_i$ denote the coordinates of element $i$; $X$ and $Y$ represent the mean centers of all elements; and $n$ stands for the total number of elements.

### 2.2.3. Kernel Density Estimation

KDE is used to identify the spatial agglomeration areas of key villages in rural tourism in which a large value of $f(x)$ indicates a dense village and a high probability of distribution [33]. The calculation formula is presented in Equation (4):

$$f(x) = \frac{1}{nh}\sum_{i=1}^{n} k\left(\frac{x - x_i}{h}\right), \tag{4}$$

where $k\left(\frac{x-x_i}{h}\right)$ is the kernel function; $(x - x_i)$ denotes the distance from the estimated point $x$ to the event $x_i$; and $h$ is the search bandwidth.

### 2.2.4. Spatial Autocorrelation Analysis

Spatial autocorrelation analysis is frequently used to reflect the degree of association between natural and social elements in space [33]. This study selects *global Moran's I* index to identify the aggregation relationship between key villages and towns of rural tourism in space, taking values in the range of [−1, 1], in which values greater than, less than, and equal to 0 indicate a positive correlation, negative correlation, and random distribution, respectively. Equation (5) presents the calculation formula as follows:

$$global\ Moran's\ I = \frac{\sum_{i=1}^{n}\sum_{j=1}^{n} W_{ij}(x_i - \overline{x})(x_j - \overline{x})}{S^2 \sum_{i=1}^{n}\sum_{j=1}^{n} W_{ij}}, \tag{5}$$

where $n$ is the number of provincial administrative regions; $W_{ij}$ denotes the spatial weight matrix; $x$ stands for the number of key villages and towns related to rural tourism in each provincial administrative region; $x_i$ and $x_j$ denote the observed values of prefecture-level cities $i$ and $j$, respectively; and $S^2$ is the variance of attribute values.

To further explore the specific locations of high- and low-value aggregation areas in the key villages and towns of rural tourism, the Getis–Ord $G_i^*$ index for key villages and towns of rural tourism was calculated using the hotspot analysis tool [28], which is calculated as follows:

$$G_i^*(d) = \sum_{j=1}^{n} W_{ij}(d)x_j \Big/ \sum_{j=1}^{n} x_j. \tag{6}$$

The parameters in Equation (6) are the same as those in Equation (5).

### 2.2.5. Geodetector

Geodetector enables the analysis of the degree of influence of each indicator on the distribution of villages and towns in rural tourism sites. $q$ has a value range of [0, 1], where large values indicate the strong influence of a factor [29,30]. Equation (7) derives the calculation formula:

$$q = 1 - \frac{1}{N\sigma^2} \sum_{h=1}^{L} N_h \sigma_h^2, \tag{7}$$

where $L$ is the spatial distribution area of tourism-focused villages and towns; $N$ and $\sigma^2$ denote the number of cells and variance of the whole, respectively, and $N_h$ and $\sigma_h^2$ refer to the number of cells and variance of layer $h$, respectively.

### 2.2.6. Multiscale Geographically Weighted Regression

The MGWR model is an improvement of the geographically weighted regression model, which adaptively adjusts the bandwidth size of variables to enable independent variables to exhibit different bandwidths; thus, it considers the scale effects of the independent variables [34]. The calculation formula is derived as follows:

$$y_i = \sum_{j=1}^{k} \beta_{bwj}(u_i, v_i)x_{ij} + \varepsilon_i, \tag{8}$$

where $x_{ij}$ is the $j$th predictor variable; $(u_i, v_i)$ is the center-of-mass coordinate for each village $i$; and $\beta_{bwj}$ is the bandwidth of the regression coefficient of the $j$th variable.

## 3. Results

### 3.1. Overall Pattern of Spatial Distribution

The overall distribution pattern of key villages and towns belonging to rural tourism in China shows a clustering pattern, with a large number of them located in eastern China and a small number in western China (Figure 1). The nearest neighbor index analysis illustrates that the actual nearest neighbor distance of key villages and towns is 10.93 km, the theoretical nearest neighbor distance is 18.27 km, the nearest neighbor index is 0.41, the $R$-value is less than one, the Z score is $-45.73$, and the significance level is $p = 0.00$. These results show that the key villages and towns of rural tourism as a whole show a clustered distribution. In terms of geographical differences, the number of key villages and towns on the east side was significantly larger than that on the west side, with the Hu line as the boundary. The population base and settlement characteristics directly influence the formation of the spatial distribution patterns of rural tourism and key villages and towns, which are greater in number in eastern China and sparse in western China.

Furthermore, this study used the Voronoi polygon CV to test for potential errors in the nearest point index. In the Voronoi polygon, a larger CV indicates that the cohesive distribution characteristics of the elements are more significant. The calculated Voronoi polygon CV for the key villages and towns of rural tourism nationwide is 97.79%, which is much larger than the cohesive threshold value of 64%. The verification results indicate that the cohesive distribution characteristics of the key villages and towns of rural tourism are more typical at a national level (Figure 2).

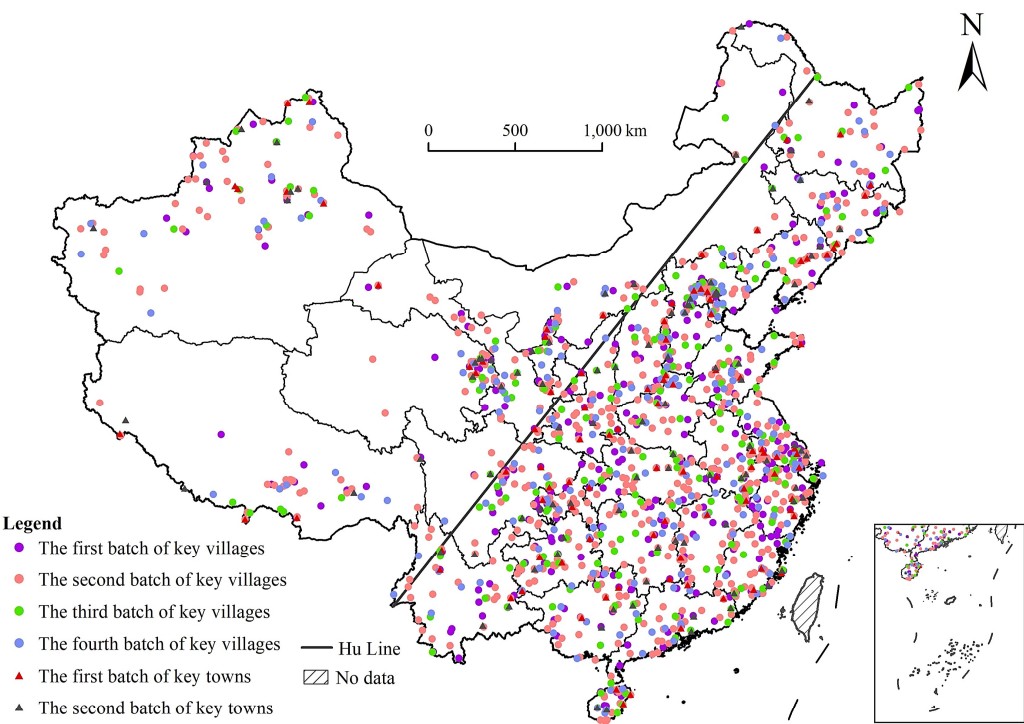

**Figure 1.** Spatial distribution of key villages and towns of rural tourism in China.

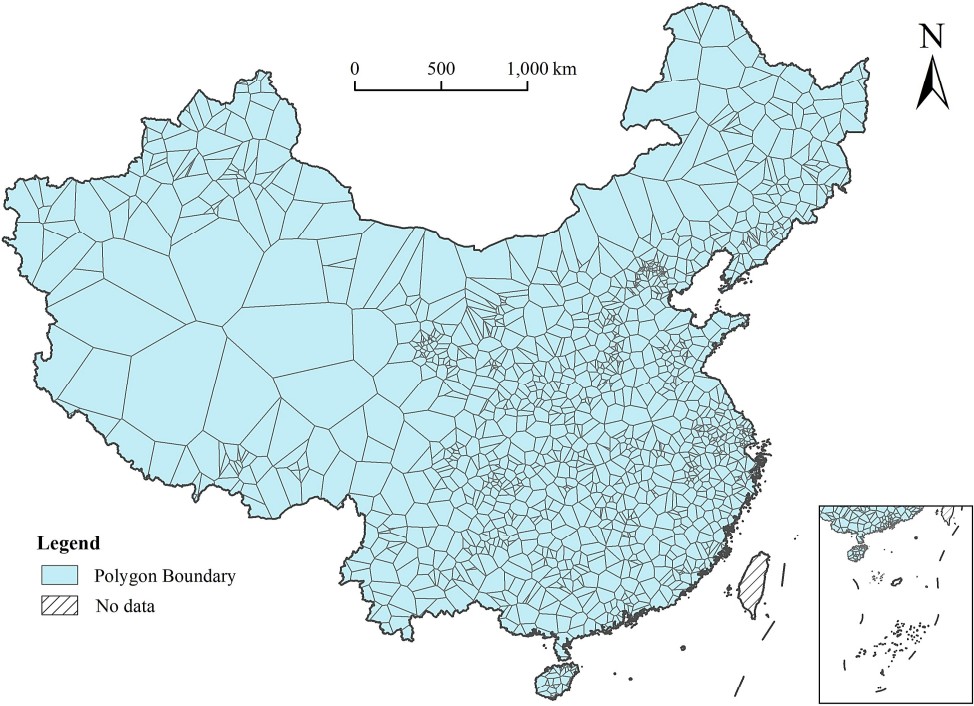

**Figure 2.** Voronoi polygon verification.

### 3.2. Center of Gravity of Spatial Distribution

The SDE can be used to reveal the overall directional characteristics of the spatial distribution of key villages and towns of rural tourism in China. In terms of the spatiotemporal distribution center of gravity (Figure 3), the average center of distribution of key villages and towns changed between 109.574° E–110.518° E and 34.056° N–34.609° N, which is roughly located at the junction between the Shaanxi and Henan provinces. In terms of a moving trajectory, the center of gravity of the spatial distribution of key villages

and towns generally moved in the northwest direction, and the distance moving in the east-west direction was greater than in the north-south direction. Moreover, the SDE area corresponding to the key villages and towns of rural tourism had a gradually increasing trend, reflecting the gradual expansion of its range of agglomeration.

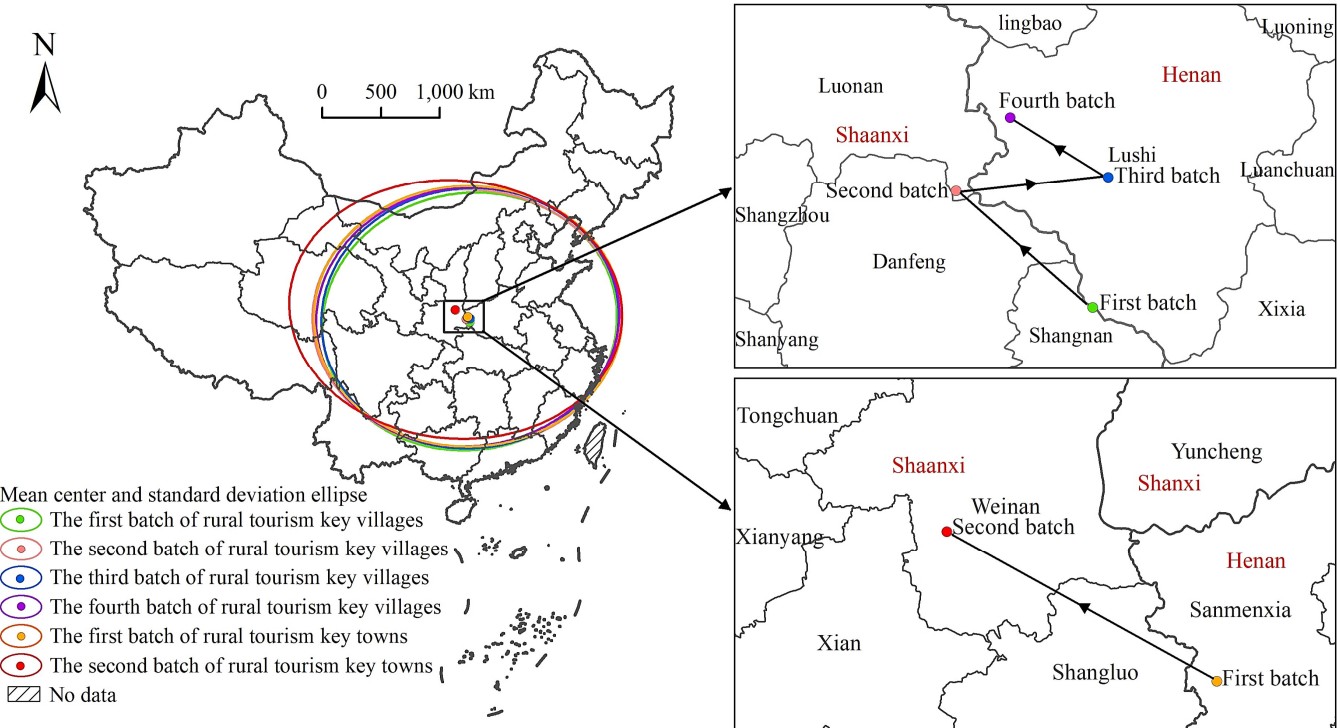

**Figure 3.** Standard difference ellipses change the key villages and towns of rural tourism.

### 3.3. Density Characteristics of the Spatial Distribution

This study used the visualization of kernel density to illustrate the evolutionary process and spatial clustering distribution characteristics of the study area further (Figure 4). The overall spatial distribution pattern displays an evolutionary trend of a band-like-point-like distribution, and the degree of double cores is becoming increasingly significant. The first batch of key villages (Figure 4a) presents the point and belt characteristics in space of two-point high-value, high-density core areas, namely, Beijing, Tianjin, Hebei, and the Bohai Sea and Shanghai, Jiangsu, Zhejiang, and the Yangtze River Delta. The first ribbon gathering area primarily comprises Beijing, Tianjin, Hebei, Shandong, Henan, Shanxi, Shaanxi, Chongqing, Sichuan, and Guizhou, which include 10 provinces and cities that form a Y-type distribution. The second ribbon agglomeration is distributed in the Yangtze River Delta region and the intersection of Anhui, Jiangxi, Hubei, and Hunan. Two more high-value core areas are observed at the intersection of two provincial capitals, namely, Qinghai and Gansu, and the eastern part of Hainan Island. In the second to fourth batches of key villages (Figure 4b–d), this ribbon is less evident with a point double-core enhancement. As a whole, the key towns (Figure 4e) maintain the same double-core distribution pattern as key villages, composed of the Beijing–Tianjin–Hebei and Yangtze River Delta regions. The band feature is less significant than that of key villages, reinforcing the overall distribution pattern (Figure 4f). These two regions, namely Beijing, Tianjin, and Hebei and Suzhou, Shanghai, and Zhejiang, have become core regions for the development of rural tourism in China, and the highly developed rural tourism in these regions has formed a strong convergence effect in central-eastern China [35].

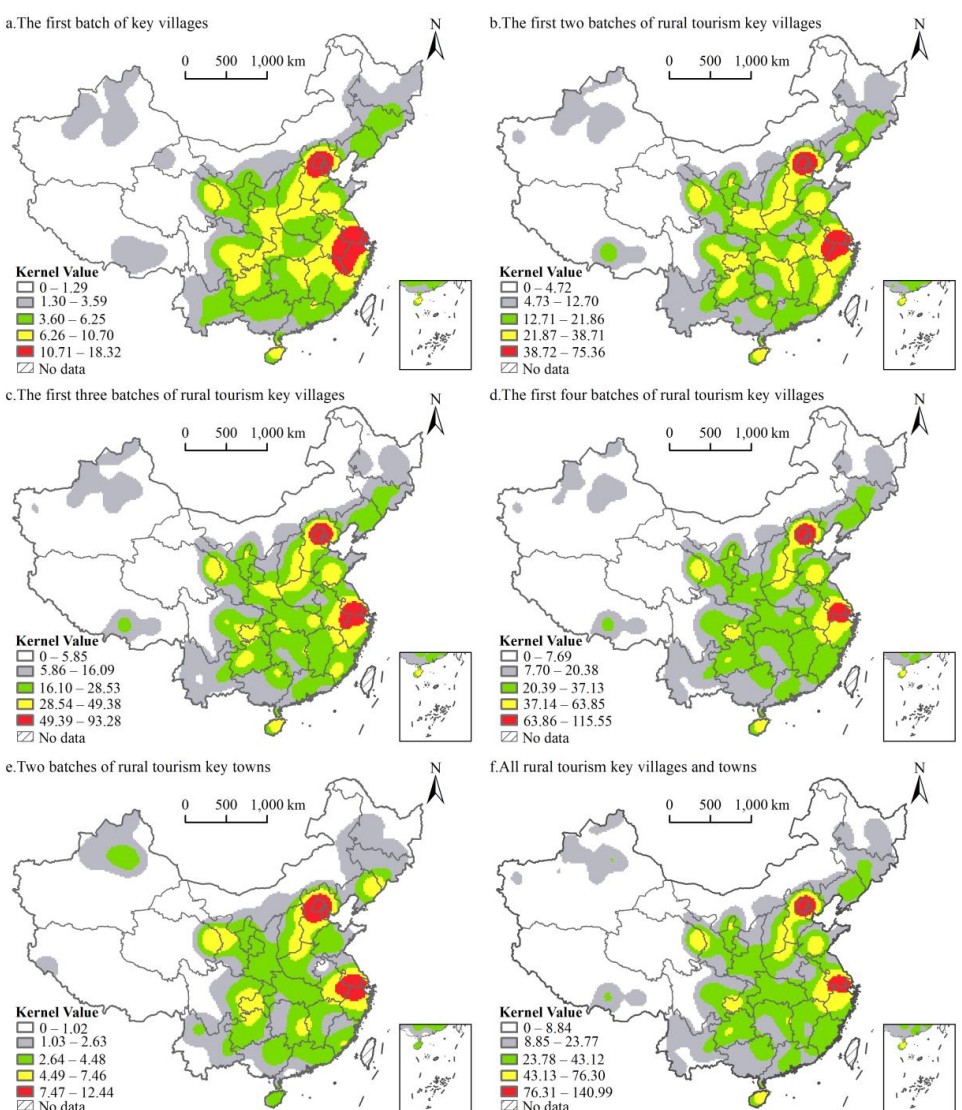

**Figure 4.** Distribution of the nuclear density of key villages and towns for rural tourism in China.

### 3.4. Distribution Characteristics of Cold Hot Spots

This paper analyzes the spatial autocorrelation of the key villages and towns of rural tourism using ArcGIS10.7, and the results demonstrate that the *global Moran's I* estimate of the spatial distribution of key villages and towns is 0.30, the normal distribution statistic Z-value is 3.21, and $p = 0.001$ with a confidence interval of 99%. This implies that the spatial distribution of key villages and towns of rural tourism in China has a significantly positive spatial correlation in a global context. In other words, areas with similar density distributions of key villages and towns tend to be close to each other in space. The neighboring spatial units are no longer independent of each other due to interdependence, and the spatial distribution in different areas is not homogeneous; there are areas that more or less tend to cluster in space.

On the basis of presenting the similarity or dissimilarity of spatial data from different regions, this study further explored regions with statistically significant clustering within each province using the local association index Getis–Ord Gi*. This study also employed the Jenks natural break method to classify the spatial layout of key villages and towns into hot (Beijing, Tianjin, and Shanghai), sub-hot (Jiangsu, Zhejiang, Gansu, Chongqing, and Hainan provinces and cities), nonsignificant, sub-cold, and cold (eight provinces and regions, such as Xinjiang, Tibet, Qinghai, and Inner Mongolia, among others) spots (Figure 5). The key villages and towns of rural tourism are clustered in low-value spaces. The sub-cold spot

area is concentrated in Jilin, Hebei, Shanxi, Shaanxi, and the seven other provinces and regions. The pattern of interlaced and uneven distribution of cold and hot areas is notable, concentrating the spatial distribution characteristics of scattered hot spots and concentrated cold spots. The Beijing–Tianjin and the Yangtze River Delta regions are economically developed regions in China with dense populations, such that key villages and towns in these areas can easily form a cluster state. Alternatively, the Qinghai–Tibet Plateau and the northwest region lag behind in the development of rural tourism due to backward economic development, a low population density, and inconvenient external transportation.

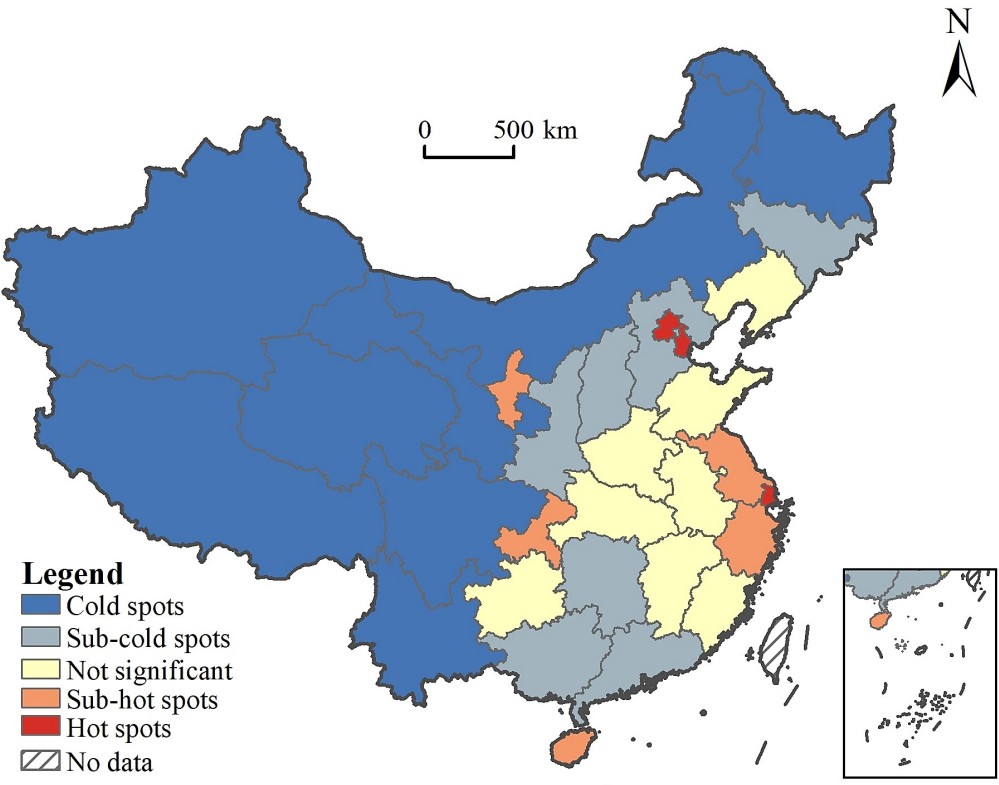

**Figure 5.** Spatial differentiation between cold and hot spots in key villages and towns of rural tourism in China.

*3.5. Analysis of Influencing Factors*

3.5.1. Geographic Detectors

The spatial differentiation of key villages and towns of rural tourism in China is the result of multiple factors. Physical geography is the basic condition for the distribution of key villages and towns and affects their overall pattern, while socioeconomic and ecological environments play an important role and affect the differences in their local distribution. This study combined the spatial differentiation characteristics of key villages and towns in rural tourism and their existing research results [5,6], considering the scientificity, correlation, and availability of index data, and constructs an index system using five dimensions, namely, natural, socioeconomic, and ecological factors, transportation support, and tourism resources. The dependent variable (*Y*) is the nuclear density value of the key villages and towns in each province (district and city) with 15 independent variable indicators ($X_i$). Among these variables, altitude, slope, temperature, and precipitation are the average data of key tourism villages and towns in each province (district and city). This study used the distance from rivers and traffic access to calculate the straight-line distances between villages and the nearest rivers and main roads. Detection indicators, such as resident population, population density, gross domestic product (GDP) per capita, and local general public budget expenditure, were adopted from the relevant data of each province (district

and city). The obtained data of each indicator were assigned to five levels in ArcGIS using the natural interruption point method to convert them from numerical to type quantities.

Table 1 presents the results of the spatial distribution of the influencing factor detection for key villages and towns. Based on the *q*-value mean ranking, the top 5 of the 15 indicators belonging to the five dimensions are population density (0.95), local general public budget expenditure (0.76), GDP per capita (0.75), road mileage (0.72), and altitude (0.34).

**Table 1.** Results of geographic exploration of factors influencing the spatial distribution of key villages and towns of rural tourism.

| Influencing Factors | Factor (Unit) | *p* | *q* | Sort |
|---|---|---|---|---|
| Natural Factors | Altitude (m) | 0.02 | 0.34 | 5 |
| | Slope (°) | 0.23 | 0.51 | |
| | Temperature (°C) | 0.23 | 0.24 | |
| | Precipitation (mm) | 0.27 | 0.21 | |
| | Distance from river (km) | 0.59 | 0.13 | |
| Socioeconomic Factors | Resident population (People) | 0.49 | 0.14 | |
| | Population density (People/km$^2$) | 0.00 | 0.95 | 1 |
| | GDP per capita (Yuan) | 0.00 | 0.75 | 3 |
| | Local general public budget expenditure (10,000 yuan) | 0.00 | 0.76 | 2 |
| Transportation | Road mileage (km) | 0.00 | 0.72 | 4 |
| | Transportation Access Distance (km) | 0.17 | 0.26 | |
| Tourism Resources | Number of A-class scenic spots (pcs) | 0.16 | 0.27 | |
| | Intangible Cultural Heritage (pcs) | 0.71 | 0.08 | |
| Ecological factors | Forest cover (%) | 0.59 | 0.11 | |
| | Wetland area (km$^2$) | 0.06 | 0.32 | 6 |

Note: Only the explanatory power of those independent variables whose *q*-values passed the significance test was ranked.

3.5.2. Spatial Variation in Influencing Factors Based on MGWR

Furthermore, to explore the spatially different characteristics of the factors that evolved the spatial pattern of tourism-focused villages and towns, this paper selected the MGWR model. The study took the nuclear density value of tourism-focused villages and towns as the dependent variable, selected the top five influencing factors in terms of their explanatory power as independent variables, and explored spatial differences in the direction and intensity of the effects of the five factors in different units of analysis. These results demonstrate that the R$^2$ of the model was 0.792, the corrected R$^2$ was 0.740, the residual sum of squares was 60.899, and the AICc was 521.223. In other words, the model passed the diagnosis of multiple covariances, and goodness-of-fit was high, reflecting the fact that the results of the geographic detector were credible. The regression coefficients of each influencing factor in each analysis unit were counted (Table 2), and their mean, standard deviation, and minimum, median, and maximum values were obtained. Intuitively, one could infer that each influencing factor displayed heterogeneity in the spatial distribution of the key villages and towns of rural tourism. Given the availability of data, the study area in this section comprises 293 prefecture-level and above cities in mainland China (excluding Hong Kong, Macao, Taiwan, Sansha City, and prefecture-level cities without the distribution of tourism-focused villages and towns).

When considering the extent to which the effects of different independent variables on the dependent variable vary with the spatial scale, the main advantage of the MGWR model is that it enables not only spatial variation in the parameter estimates but also the generation of unique optimal bandwidths for the relationship between the dependent variable and each independent variable. As obtained from Table 2, the bandwidths (action scales) of different variables varied widely, with the smallest bandwidth being 43 for altitude and

road mileage, followed by GDP per capita and population density with bandwidths of 52 and 68, respectively. This indicates that the distribution of key villages and towns for rural tourism varies spatially with changes to the distribution of these factors. The bandwidth of local general public budget expenditures is the largest at 292 with relatively little spatial heterogeneity. Evidently, the effects of the respective variables are spatially non-stationary; however, the degree of variability and the characteristics presented vary. Moreover, the results of each factor are visually expressed by the natural breakpoint grading method (divided into five levels; Figure 6).

**Table 2.** Statistical description of the MGWR model regression coefficients of factors influencing the spatial distribution of tourism-focused villages and towns.

| Factor | Bandwidth | $p$ | Mean | STD | Min | Median | Max |
|---|---|---|---|---|---|---|---|
| Intercept | 43 | 0.000 | 0.417 | 0.423 | −0.301 | 0.462 | 1.307 |
| Altitude | 43 | 0.000 | 0.635 | 0.455 | −0.205 | 0.609 | 1.657 |
| Population density | 68 | 0.000 | 0.430 | 0.399 | −0.048 | 0.316 | 1.286 |
| GDP per capita | 52 | 0.041 | 0.186 | 0.188 | −0.279 | 0.183 | 0.632 |
| Local general public budget expenditure | 292 | 0.065 | −0.053 | 0.006 | −0.061 | −0.054 | −0.018 |
| Road mileage | 43 | 0.037 | 0.008 | 0.265 | −0.703 | 0.006 | 0.678 |

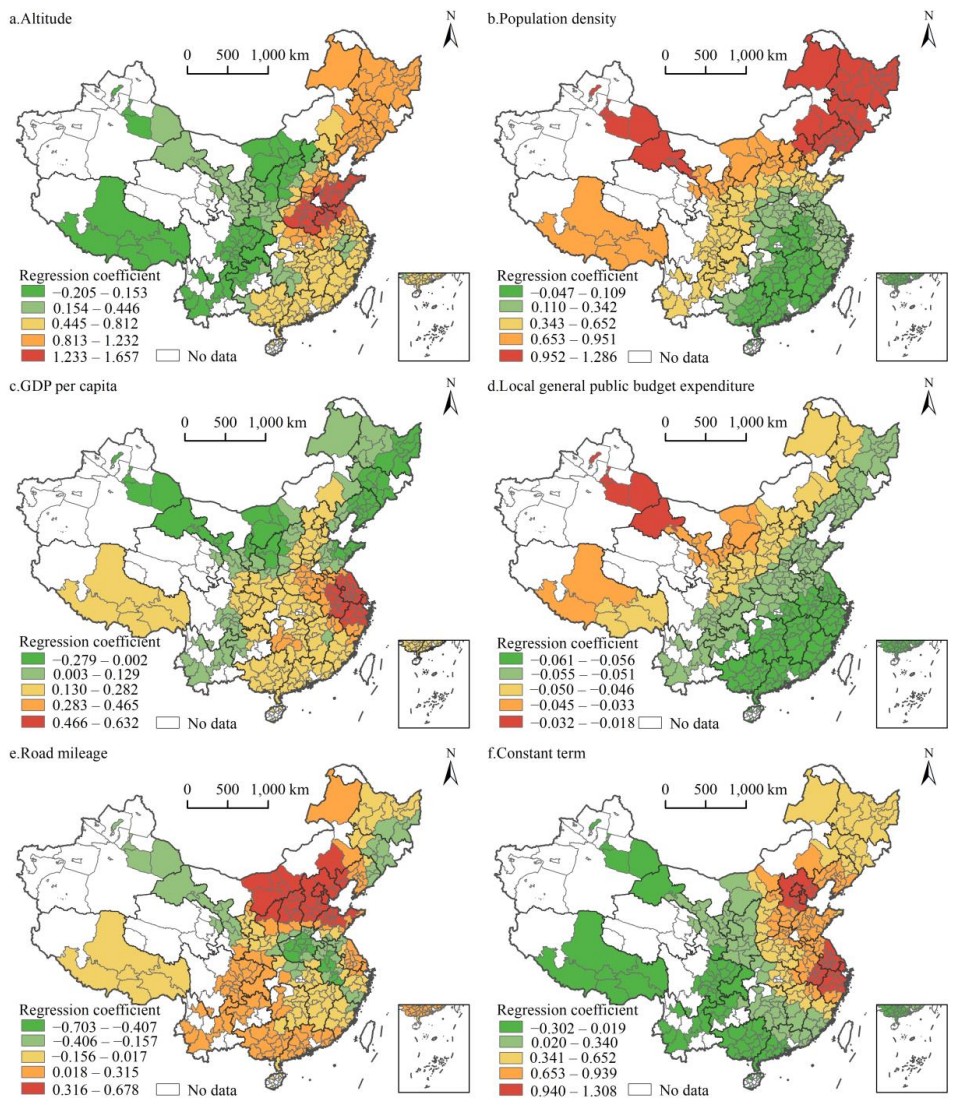

**Figure 6.** Spatial distribution of regression coefficients of influencing factors in MGWR model.

(1)   Altitude

The spatial differences in the effects of altitude (regression coefficients ranging from −0.205 to 1.657) were significant, and the number of analysis units with regression coefficients greater than 0 accounted for more than 90% of the total, with positive effects predominating. On the whole (Figure 6a), the influence of altitude on the distribution of key villages and towns in rural tourism was characterized as "high in the east and low in the west". The distribution of key villages and towns of rural tourism in eastern China, especially in the intersection of four provinces, namely, Shandong, Henan, Jiangsu, and Anhui, was influenced by altitude and concentrated in high-altitude areas. A possible reason for this is that villages and towns in low-altitude areas have a plain landscape, and villages and towns in these areas are primarily agricultural and lack tourism resources. In addition, the villages and towns in plain areas are more homogeneous in terms of tourism resources and landscape, which are difficult to develop. Meanwhile, in western China, especially within the first and second steps of the ladder, the higher the altitude, the lower the distribution of key villages and towns for rural tourism. This may be because the high-altitude terrain and complex topography cluster the population and villages on the sides of erosion-type canyons and basin edges, such that the regression coefficient of altitude is lower. Notably, however, the altitude here is only relative, and the calculation found that key tourism villages and towns distributed within 1000 m above sea level accounted for 81.57% of the analysis units, which shows that most of the villages are still distributed in low-altitude areas.

(2)   Population density

Population density (regression coefficients ranging from −0.047 to 1.286) was positively correlated with the distribution of key villages and towns of rural tourism. Figure 6b suggests that the spatial distribution of the regression coefficients of population density significantly varied with an overall "high in the north and low in the south" distribution, which is a particularly prominent imbalance. The regression coefficients were positively correlated for 89.08% of the analyzed units. In particular, in the three northeastern provinces and northern Inner Mongolia, northwestern Gansu, northern Qinghai bordering areas, and eastern Xinjiang, which present high population densities and large distributions of key villages and towns of rural tourism, the analysis units that are negatively correlated were concentrated in central and southern China. On the whole, the construction of key villages and towns in northwest China was more dependent on population distribution.

(3)   GDP per capita

The regression coefficient of GDP per capita (−0.279 to 0.632) was positively correlated in 83.28% of the units analyzed, and the difference between the maximum and minimum values of the regression coefficient was 0.911, indicating that the purchasing power of residents widely varied across regions. Figure 6c indicates that the regression coefficients were larger in Central and South China and Tibet but were largest in the middle and lower reaches of the Yangtze River, such as Shanghai, Jiangsu, and Zhejiang. In other words, the higher the GDP per capita, the stronger the willingness to travel and the purchasing power of tourism, and the more direct the impact on the distribution of tourism-focused villages and towns. This implies that the high level of regional economic development, the high potential of the source market, and the strong demand for tourism can promote the construction of rural tourism in key villages and towns.

(4)   Local general public budget expenditure

The regression coefficient of the local general public budget expenditure and its fluctuation is much smaller than those of the four other independent variables, at only −0.061 to −0.018, and this effect is only weakly different in space. The influence of the local general public budget expenditure on the development of rural tourism in key villages and towns is negative in all units of analysis. The constraint effect is most prominent in the area east of the Hu line. To summarize, the influence on the distribution of key villages

and towns that rural tourism exhibits causes a decreasing trend from the northwest to the southeast with an increase in the local general public budget expenditure. Thus, an increase in local general public budget expenditure exerts an inhibitory effect on the concentration of key villages and towns of rural tourism, and villages are more likely to be distributed in regions with relatively backward economic development.

(5)  Road mileage

The fluctuations in the regression coefficients of road miles traveled were significant ($-0.703$ to $0.678$), and the spatial heterogeneity of the effects was strong. The number of analysis units exhibiting positive and negative effects was 148 and 145, respectively. In the southwestern and central regions of Shanxi, Hebei, Shandong, and Inner Mongolia, increasing the mileage of roads could effectively promote the development of rural tourism in key villages and towns. Moreover, special attention could be given to the supporting role of transportation networks. In these regions, convenient transportation provides basic conditions for the movement of people, material circulation, and information transfer between villages and cities, which becomes an important factor influencing the distribution of rural tourism in key villages and towns. At the same time, the effects of convenient transportation were negative in several units of analysis, which was mainly related to the mismatch between the density of road networks in these areas and the number of key villages and towns of rural tourism nationwide under the influence of policies and other factors.

From the results of the analysis of the dominant factors mentioned above, it is easy to see that the spatial differentiation of key villages and towns in rural tourism in China is indeed the result of the joint action of multiple factors. According to the results of a previous study, the spatial distribution characteristics, location conditions, and tourism sustainability-related contents of key villages and towns related to rural tourism in the four regions of eastern, northeastern, central, and western China are summarized in Table 3.

**Table 3.** Statistics of key villages and towns in rural tourism for the four major regions of China by sub-district.

| | Spatial Distribution Characteristics | Location Conditions | Tourism Sustainability Case |
|---|---|---|---|
| Eastern China | Centralized distribution with Beijing-Tianjin-Hebei and the Yangtze River Delta as the dual correct | Eastern China has obvious location advantages: low altitude, convenient transportation, and tourism products focusing on vacation and leisure. | Nanjing's Fuzimiao and Jiangnan Gongyuan are both famous cultural attractions that focus on tourism sustainability while preserving traditional culture. |
| Northeast China | More in the south, less in the north | It is a frontier region with relatively poor natural conditions and infrastructure and a low level of economic development. | Changchun Cinema City is a comprehensive venue that includes facilities such as a movie theater, an amusement park, and a shopping mall, combining culture, entertainment, and shopping. |
| Central China | Concentrated and continuous distribution in a "gong" shape | Most of the provinces in the central region of China are either highly populated or have large tourism resources, attracting more tourists to visit. | The Yellow Crane Tower in Hubei is one of the representatives of Jiangnan culture, which blends with natural mountains and forests to create a unique urban landscape garden. |
| Western China | Based on "provincial capital cities and transportation arteries," centralized and continuous distribution | Western China is rich in tourism resources, but transportation conditions, infrastructure, and human capital need to be improved. | Jiuzhaigou in Sichuan is a famous nature reserve that is not only a tourist attraction but also an important ecological reserve. |

## 4. Discussion

This study analyzes the spatial distribution and characteristics of key villages and towns of rural tourism nationwide at the macroscopic level and reveals the intensity of factors affecting the spatial distribution of key villages and towns in rural tourism. We also examined the spatial differences of influencing factors, which present practical significance for the sustainable development of rural tourism, the construction of villages, and the promotion of the comprehensive revitalization of villages. However, the development of rural tourism is an interdisciplinary and cross-field industry involving many fields, such as tourism, agriculture, culture, commerce, and trade. This study mainly focuses on the macro perspective of the current situation of tourism development in key villages and towns of rural tourism in China, which is one-sided and limited due to the limitations of channels and sources of information acquisition and the limited level of personal research abilities. At the same time, the layout of key villages and towns for rural tourism is affected by a combination of natural, humanistic, and other factors. Although this study used the MGWR model to identify the spatially divergent characteristics of factors that influenced the distribution of key villages and towns for rural tourism in China, it has several shortcomings. First, the factors influencing the distribution of key villages and towns in rural tourism are complex topics, and the selection of some influencing factors might be unreasonable due to limited data availability. Second, the interpretation of the scale effect is a difficult problem in the application of the MGWR model. Although this study endeavored to explain the scale effect, the lack of strong support led to a weak discussion regarding this. Third, this study is limited to spatial heterogeneity using cross-sectional data; the factors that influence the key villages and towns of rural tourism are not only spatially heterogeneous and exhibit scale effects but could also be temporally heterogeneous. This aspect requires further exploration to improve the diversity of the time series data and optimize the model. As a demonstration and as typical of the utilization and development of rural tourism, key villages, and towns are important carriers for the promotion and flourishing development of rural tourism and can guide its benign and healthy development. In future studies, the focus should be on the research and construction support of the comprehensive effect of key villages and towns in rural tourism. In particular, advanced methods such as multidimensional data collection, intelligent analysis, and visualization processing should be used to conduct empirical research on key villages and towns in rural tourism at multiple levels and from multiple perspectives. This could help to achieve integrated urban-rural development and rural transformation, reconstruction, and revitalization with a view to better promote the sustainable development of rural tourism in China.

## 5. Conclusions

This paper takes the key villages and towns of rural tourism in China as the research object, analyzes their spatial distribution patterns and clustering characteristics using methods such as ArcGIS spatial analysis, and explores the factors that influence their spatial patterns and spatially divergent characteristics using the geographic probe and the MGWR model. This study draws the following conclusions:

1. The key villages and towns of rural tourism in China generally display a clustering distribution pattern of more in the east and less in the west, and the center of gravity of spatial distribution generally moves to the northwest, indicating a trend of a belt-like-point-like distribution. Thus, spatial distribution demonstrates a significant positive correlation and tends to be unbalanced; as such, hot spots are only scattered in the Beijing–Tianjin–Hebei and the Yangtze River Delta regions, and cold spots are concentrated and contiguous in the northwest.

2. According to the geographic detection results, this study found that the explanatory degree of each influencing factor on the spatial differentiation of key villages and towns of rural tourism presents significant differences. Among them, population density, resident consumption expenditure, GDP per capita, road mileage, and altitude are the major factors that influence the spatial distribution of key villages and

towns of rural tourism. This spatial differentiation is majorly influenced by population, economic development, and transportation location and less by natural factors, such as topography and river systems, which are the result of a combined effect of multiple factors.

3. The MGWR model calculations indicate that the key villages and towns of rural tourism are concentrated in low-altitude areas, and their degree of influence points to a decreasing distribution feature from east to west. Furthermore, the construction of key villages and towns in the northwest and north regions is more dependent on population compared to that of other regions. The constraint effect of local general public budget expenditure is most prominent in the area east of the Hu line, and GDP per capita significantly and positively influences the spatial distribution, which, in turn, is significantly and positively influenced by GDP per capita. The enhancement effect of road mileage is prominent in the majority of regions, which should be given higher attention in the construction and development of key villages and towns related to rural tourism to maximize their supporting and promoting role.

**Author Contributions:** Conceptualization, Q.Z. and J.S.; methodology, Q.Z.; software, Q.Z.; validation, Q.Z.; formal analysis, Q.Z.; resources, Q.Z. and J.S.; data curation, Q.Z.; writing—original draft preparation, Q.Z.; writing—review and editing, Q.Z.; visualization, Q.Z.; supervision, J.S.; project administration, J.S., J.L., J.C. and X.K.; funding acquisition, J.S., J.L., J.C. and X.K. All authors have read and agreed to the published version of the manuscript.

**Funding:** This research was funded by the National Natural Science Foundation of China (41961031, 42271228, 42361028), the Guizhou Provincial Science and Technology Project (Qiankehe Foundation-ZK (2022), General 313), and Guizhou Normal University, 2019 Academic New Seedling Cultivation and Innovation Exploration Special Project (Qian Shi Xin Miao (2019), No. A13).

**Institutional Review Board Statement:** Informed consent was obtained from all subjects involved in this study.

**Informed Consent Statement:** Not applicable.

**Data Availability Statement:** Not applicable.

**Conflicts of Interest:** We declare that we have no financial and personal relationships with other people or organizations that could interfere with our study.

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
