# Peer review of "Spatial Patterns of Key Villages and Towns of Rural Tourism in China and Their Influencing Factors"

_sustainability, doi:10.3390/su151813330_

Round 1

Reviewer 1 Report

The authors have prepared a good article and it is significant from a geographical point of view, but important points should be considered in this regard.

1- Published articles based on spatial analysis have increased. Innovation is not seen in this article.

2- In the general introduction, the lack of problems and emphasis on specific and practical goals is not observed.

3- General discussions and conclusions without specific and effective results on the production of science are not seen.

4- The article has no achievements at the international level and its results can be used more at the regional or national level.

5- According to the issues raised, it is suggested that this article be published in scientific journals of China or Southeast Asian journals.

Reviewer 2 Report

Dear Authors,

1. What is the main question addressed by the research?
TOURISM IN VILLAGES IN CHINA
2. Do you consider the topic original or relevant in the field? Does it
address a specific gap in the field?
YES. IT IS QUITE INNOVATIVE
3. What does it add to the subject area compared with other published
material?
IT IS QUITE INNOVATIVE. THERE ARE NOT MANY STUDIES IN THE AREA
4. What specific improvements should the authors consider regarding the
methodology? What further controls should be considered?
I THINK THE METHODOLOGY IS APPROPRIATE
5. Are the conclusions consistent with the evidence and arguments presented
and do they address the main question posed?
YES
6. Are the references appropriate?
YES
7. Please include any additional comments on the tables and figures.
NOT REQUIRED

I have read your article and found it to be quite interesting. I have a few comments:

1. The discussion section should be expanded and more detailed.

2. Please add a limitations and further research section, after the conclusions.

Reviewer 3 Report

The research is proposed as an exhaustive territorial and geographical analysis of the towns and cities that the Chinese authorities identify as key for rural tourism from a sample of more than 1,500 municipalities. This is the most noteworthy quality compared to other types of studies.

The methodological procedure uses various factors, variables and tools to verify and determine the characteristics of these population settlements. In addition, an attempt is made to provide useful knowledge and valid conclusions to show possible situations to be corrected that can be adopted in the criteria of the Chinese authorities in this regard. The practical and applied nature is also outstanding.

The contextualization, explanation and justification of the research is well constructed and possible deficiencies detected in the study, motivated by the lack of data, are even argued in the discussion section, and future analyzes with possible corrective measures are proposed. Likewise, the reference sources suitably respond to the purposes of the article, offering figures and tables that are sufficiently illustrative of the content.

Perhaps it would be interesting or convenient to explain, even briefly in the introductory part, the characteristics of Chinese rural tourism supply and demand, taking into account who the tourists are (national or international profile).

It is mentioned that the impulse of this tourist modality is influenced by Covid 19, although there are other aspects that are perhaps more important. It is presented as an alternative to other more traditional types of tourism (sun and beach, cultural and urban) beginning to show symptoms of saturation and fatigue and need a reconversion or reinvention, at least in other countries.

Round 2

Reviewer 1 Report

Thanks to the authors who have done the terminology. But I need to emphasize the previous important points.

The article is still local and mostly describes the geographical features and the existing situation and has not been able to provide a framework and model for further research. Therefore, the results of the article are national and not international.

1- The article is not within the scope of the Journal's goals and has not been able to bring results that match it.

2- Although the rural tourism development model has been emphasized as the goal of the article in the recent edition, its results and the reforms carried out are still national and local in nature and cannot be used internationally and have not been able to introduce a framework for the model in question.

3- The article has not been able to introduce a model for sustainable development for future work in international dimensions.

4- The amendments made have not been able to change much in the article and still emphasize the description of the current situation.
